# Comparative effectiveness and safety of insulin reference biologics versus biosimilars for types 1 and 2 diabetes mellitus: Protocol for a systematic review of real-world studies

Martin K. H. Ho[1]*, Araniy Santhireswaran[1], Tara Gomes[1,2,3], Muhammad Mamdani[1,2,3], Mina Tadrous[1,2,4]

**1** Leslie Dan Faculty of Pharmacy, University of Toronto, Toronto, Ontario, Canada, **2** Institute of Health Policy, Management and Evaluation, University of Toronto, Toronto, Ontario, Canada, **3** Li Ka Shing Knowledge Institute, St. Michael's Hospital, Unity Health Toronto, Toronto, Ontario, Canada, **4** Women's College Hospital, Toronto, Ontario, Canada

* martinkh.ho@mail.utoronto.ca

## Abstract

### Introduction

Diabetes mellitus is characterized by insulin deficiency or resistance. The two main types of diabetes mellitus are type 1 (T1DM) and type 2 (T2DM). Insulin is the mainstay of therapy for T1DM and often the last-line therapy for T2DM. Biosimilar insulins are cost-saving alternatives to reference products that may improve access for patients and sustainability for healthcare systems. Despite supporting evidence from randomized controlled trials, biosimilar insulin uptake is poor, and real-world evidence of their safety and effectiveness is limited.

### Objective

Our objective is to compare the real-world effectiveness and safety of insulin biosimilars versus reference products in adults with diabetes mellitus.

### Eligibility criteria

We will include observational studies and open-label pragmatic randomized controlled trials. We will exclude other randomized controlled trials, literature reviews, meta-analyses, case series, case reports, study protocols, opinion pieces, and conference abstracts. Our primary effectiveness outcome will be glycated hemoglobin (HbA1c) and our primary safety outcome will be hypoglycemia. Our secondary outcomes will include fasting plasma glucose; time in range; microvascular complications; health-related quality of life; physician visits, emergency department visits, and hospital admissions for hypoglycemia, hyperglycemia, and diabetic ketoacidosis; weight gain; immunogenicity; injection site reactions; and incident cancers.

**Data availability statement:** No datasets were generated or analysed during the current study. All relevant data from this study will be made available upon study completion.

**Funding:** The author(s) received no specific funding for this work.

**Competing interests:** The authors have declared that no competing interests exist.

**List of abbreviations:** DPP-4: dipeptidyl peptidase-4; GLP-1: glucagon-like peptide-1; GRADE: Grading of Recommendations, Assessment, Development and Evaluation; HbA1c: glycated hemoglobin; PRECIS-2: PRagmatic Explanatory Continuum Indicator Summary-2; PRISMA-P: Preferred Reporting Items for Systematic Reviews and Meta-Analyses Protocols; RoB 2: Cochrane risk-of-bias tool for randomized trials version 2; ROBINS-I V2: Risk Of Bias In Non-randomized Studies of Interventions Version 2; SGLT2: sodium-glucose transport protein 2; T1DM: type 1 diabetes mellitus; T2DM: type 2 diabetes mellitus.

## Methods

The search strategy combines three key concepts: diabetes, insulin, and biosimilars. We will conduct a structured search in MEDLINE, EMBASE, and International Pharmaceutical Abstracts. We will also search in grey literature databases, targeted websites, and the Google search engine. Finally, we will scan forward and backward citations. Articles will be screened, extracted, and appraised independently by two reviewers. Data will be descriptively summarized.

## Discussion

Our systematic review of the real-world evidence on biosimilar insulins can help support clinical and policy decisions that impact the care of patients with T1DM or T2DM.

## Introduction

Diabetes mellitus is a chronic condition where the body does not produce enough or effectively use insulin, leading to high blood glucose levels [1]. In 2021, there were over 500 million adults aged 20–79 years living with diabetes worldwide, and the incidence is increasing [1]. Among these, it is estimated that there are at least 150–200 million people who require insulin therapy [2].

There are two main types of diabetes: type 1 (T1DM) and type 2 (T2DM), with over 90% of cases being T2DM [1]. T1DM is an autoimmune disorder that destroys pancreatic beta cells, which produce insulin [1], for which insulin is the mainstay of therapy [3,4]. T2DM is caused by impaired beta cell function and insulin resistance [5]. In T2DM, the first-line therapy is metformin; second-line therapies include dipeptidyl peptidase-4 (DPP-4) inhibitors, glucagon-like peptide-1 (GLP-1) receptor agonists, sodium-glucose transport protein 2 (SGLT2) inhibitors, insulin secretagogues, and insulin [6]. Insulin is particularly useful for patients with ketosis, unexpected weight loss, or symptoms of hyperglycemia (polyuria or polydipsia) [7,8]. Otherwise, insulin is usually added when other agents are insufficient due to its risk of hypoglycemia and weight gain [6,9].

As the patents for some reference insulin products have expired, biosimilars have entered the market. The definition of 'biosimilar' varies across countries. For example, some definitions of 'biosimilars' only include products that are approved in countries with stringent regulatory frameworks for biosimilars (i.e., those that are consistent with guidance from the European Medicines Agency, United States Food and Drug Administration, or World Health Organization) [10,11]. In contrast, others also include non-innovator products that are approved without stringent regulatory frameworks and may or may not meet sufficient quality, efficacy, or safety standards [11]. Throughout this protocol, we will implement the most broad definition for 'biosimilars', referring to all of the products described above.

Biosimilars are supported by clinical studies demonstrating no meaningful difference in efficacy, safety, or immunogenicity compared to the reference products

[12,13]. Because they cost less that reference insulin products, they can improve accessibility for patients and sustainability for public healthcare systems [14,15]. However, variations in biosimilar manufacturing may impact their efficacy and safety [13,16]. This 'similar but not identical' paradigm has led to hesitation among patients and clinicians in adopting biosimilars [17]. This is reflected in the poor uptake of biosimilar insulins. In the second quarter of 2023, biosimilar insulin uptake was less than 50% of treatment days in the USA, Canada, Japan, United Kingdom, and various countries in the European Union [18].

While biosimilar insulins are supported by clinical trials as well as their systematic reviews and meta-analyses, [19–24] real-world evidence is limited. Existing biosimilar insulin reviews [25–27] that include real-world studies have only found three observational studies on this topic [28–30]. However, these systematic reviews did not cover both types of diabetes or all currently available biosimilar insulins. We have summarized the differences between these past reviews and our planned review in S1 Appendix. Furthermore, these reviews only included research up to March 27, 2019, after which new biosimilars for insulin aspart and glargine have been approved in Europe, the USA, and Canada [31–33]. We anticipate additional real-world studies in the past six years, such as those by Pitlick *et al.* (2020), Pham *et al.* (2022), and AlRuthia *et al.* (2022) [34–36]. Real-world data on biosimilar insulin use can help inform clinical and policy decisions, such as public formulary coverage, and overcome barriers to uptake [15,17]. Therefore, our objective for this systematic review is to compare the real-world effectiveness and safety of insulin biosimilars versus reference products in adults with types 1 and 2 diabetes mellitus.

## Methods

We registered the systematic review protocol with the PROSPERO database (CRD42024598628). This protocol follows the Preferred Reporting Items for Systematic Review and Meta-Analysis Protocols (PRISMA-P) guidelines (see S2 Appendix) [37]. Amendments to the protocol during study conduct will be documented in the final systematic review publication. We expect data screening and extraction to be completed by November 2025 and results to be available by December 2025.

### Eligibility criteria

We will include observational studies (e.g., cohort studies, case-control studies, within-subject studies, cross-sectional studies, interrupted time series, controlled before and after studies) and open-label pragmatic randomized controlled trials (RCTs). As our objective is to evaluate insulin use in the real world, we will exclude experimental studies that are not open-label pragmatic RCTs. The PRECIS-2 tool helps distinguish between pragmatic versus explanatory RCTs [38]. However, even when using this tool, there is no simple dichotomous threshold for making this distinction; reviewers will use a combination of PRECIS-2 and their judgement. We appreciate that the PRECIS-2 authors do not recommend dichotomizing RCTs as a whole as pragmatic or explanatory [39]. Our use of PRECIS-2 to exclude explanatory RCTs is to avoid research waste through duplication of effort, as there have been systematic reviews of RCTs with similar research questions as recent as 2023, [19] while fulfilling our objective of summarizing the available real-world evidence. We will also exclude literature reviews, meta-analyses, case series, case reports, study protocols, opinion pieces, conference abstracts, and studies with unavailable full texts free of charge through the University of Toronto library. We will focus on studies involving adults (aged ≥18 years) with T1DM or T2DM for any duration who take insulin biosimilars compared to their respective reference products. We will include studies of all follow-up durations.

Our primary effectiveness outcome is change in glycated hemoglobin (HbA1c). Our secondary effectiveness outcomes include fasting plasma glucose; time in range; microvascular complications (retinopathy, nephropathy, neuropathy); and health-related quality of life. Our primary safety outcome is hypoglycemia. Our secondary safety outcomes include physician visits, emergency department visits, and hospital admissions for hypoglycemia, hyperglycemia, and diabetic ketoacidosis; weight gain; immunogenicity; injection site reactions; and incident cancers.

We will not have restrictions on language or publication date. A draft eligibility form for screening titles and abstracts, or full-text articles, is presented in S3 Appendix.

## Information sources and literature search

We will conduct a structured search in MEDLINE, EMBASE, and International Pharmaceutical Abstracts. We will also search grey literature databases (Trip Pro, OAIster), targeted websites (see S4 Appendix; each website will be scanned using Advanced Search on Google), and the Google search engine (the first five pages for five different keyword queries will be scanned). As a supplement, we will use citationchaser to scan the reference lists of previous reviews and included articles, as well as studies that cite the included articles [40].

To develop the search strategy, we conducted a preliminary limited search in MEDLINE to identify key search concepts. Using these search concepts, we scanned the titles and abstracts of relevant articles to select subject headings and text words to develop a search strategy for MEDLINE (S5 Appendix). The subject headings, textword and keyword queries, as well as other database-specific syntax will be adapted for EMBASE and International Pharmaceutical Abstracts.

## Data management

Literature search results will be imported to the reference manager EndNote 20 (Clarivate Analytics, Pennsylvania, USA) and the review manager Covidence (Veritas Health Innovation, Melbourne, Australia) for deduplication [41,42]. Records with missing titles or abstracts will proceed directly to the full-text screening phase.

## Study selection process

The draft screening criteria for titles and abstracts (S3 Appendix) will be pilot tested on a random sample of 5% of the articles from the literature search. Two reviewers will screen the titles and abstracts. At this step, open-label RCTs that are potentially relevant will be included; whether they are pragmatic will be determined in the full text screening step. Any disagreements will be discussed with the team and the eligibility criteria will be revised as necessary. This process will be repeated on a new sample of citations until adequate inter-rater reliability (Cohen's kappa >0.60) is reached [43]. Two reviewers will then independently screen all the literature search results. Disagreements will be resolved by discussion or a third reviewer.

Similarly, the draft screening criteria for full-text articles (S3 Appendix), will be pilot tested. The full-text articles will then be independently screened by two reviewers. For open-label pragmatic RCTs, reviewers will use the PRECIS-2 tool to help decide whether they are pragmatic [38,39]. Trials that score 4 or 5 out of 5 across many domains are more likely to be pragmatic RCTs. Domains with high scoring variation by reviewers (e.g., by 2 levels or more) will be resolved by discussion or a third reviewer. Disagreements on the overall pragmatism of studies will also be resolved by discussion or a third reviewer. The PRECIS-2 ratings for candidate RCTs, including those that were ultimately excluded, will be published with the completed systematic review.

## Data items and data collection process

Data will be extracted on the following:

• Study characteristics: journal, funding sources, country, study design, study period, length of follow-up, setting, data sources, data type, type of insulin, biosimilar, reference product, delivery system, dosage;

• Patient characteristics: number of patients, age (mean and standard deviation), sex, type of diabetes, duration of diabetes, baseline HbA1c; and

• Outcome results (at 6, 12, 24 months, and the longest duration of follow-up): change in HbA1c, fasting plasma glucose, time in range, microvascular complications, health-related quality of life, physician visits, emergency department visits, hospital admissions, weight gain, immunogenicity, injection site reactions, incident cancers.

 

The draft data collection form (S6 Appendix) will be pilot tested on a random sample of five to ten included studies. The form will be revised as necessary, until adequate inter-rater reliability (Cohen's kappa >0.60) is reached. Next, two reviewers will independently extract data from all included studies. Disagreements will be resolved by team discussion.

## Methodological quality/risk of bias appraisal

Two reviewers will independently appraise the risk of bias of the included observational studies using the Risk Of Bias In Non-randomized Studies of Interventions Version 2 (ROBINS-I V2) tool, [44] and of the included pragmatic randomized controlled trials using the revised Cochrane Risk of Bias tool for randomized trials (RoB 2) [45]. Disagreements will be resolved by discussion or a third reviewer. The certainty of evidence will be assessed using the Grading of Recommendations, Assessment, Development and Evaluation (GRADE) tool [46].

## Synthesis of included studies

We will summarize the study characteristics, patient characteristics, outcome results, and risk of bias of the included studies. We do not plan to conduct a meta-analysis because our systematic review only includes observational studies and open-label pragmatic RCTs. Therefore, we anticipate a high level of heterogeneity in study design (comparators, outcomes, user definitions, bias, etc.) that would preclude a valid pooling of results. To qualitatively assess this heterogeneity, we will present the above findings in a summary table, organized by study design. A draft summary table is presented in S7 Appendix. If feasible, we will explore the results by the type of diabetes and insulin.

## Significance

Diabetes mellitus affects more than 500 million people globally, and this number is growing annually [1]. Insulin is first-line therapy for T1DM [3,4] and the bulwark last-line therapy for T2DM [6,9]. While insulin reference products are costly, their biosimilars cost less and can improve access for patients.

A previous systematic review identified three categories of barriers and enablers of biosimilar uptake: the system, healthcare providers, and patients [17]. The most frequently discussed barrier for the system was a lack of effective policies or guidelines, for healthcare providers was a lack of awareness, and for patients was concern about safety and efficacy [17]. Countries have explored different strategies to encourage biosimilar uptake, such as automatic substitution and mandatory non-medical switching [47]. This has had some success. For example, in 2019, the Canadian province of British Columbia implemented a mandatory non-medical switch policy for insulin glargine, where only the cost of the biosimilar would continue to be covered [48]. The uptake of biosimilar insulin glargine rose from less than 20% to 99.6% [48]. However, the global uptake of biosimilar insulins remains poor, with less than 50% of treatment days covered in various developed countries in 2023 [18].

With increasing use of biosimilar insulins, there is an opportunity for real-world studies to compare their effectiveness and safety to insulin reference products. Real-world studies often include large populations and diverse clinical settings, which can detect uncommon or long-term outcomes. Therefore, our systematic review of the real-world evidence can help support clinical and policy decisions that impact millions of patients with T1DM or T2DM worldwide.

## Supporting information

**S1 Appendix. Comparison with past reviews.**
(DOCX)

**S2 Appendix. PRISMA P Checklist.**
(DOCX)

**S3 Appendix. Draft eligibility form.**
(DOCX)

**S4 Appendix. Targeted website search.**
(DOCX)

**S5 Appendix. MEDLINE search strategy.**
(DOCX)

**S6 Appendix. Draft data extraction form.**
(DOCX)

**S7 Appendix. Draft summary table of included studies.**
(DOCX)

## Acknowledgments

We thank Dr. Michael Fralick for his clinical guidance in developing the PROSPERO registration for this systematic review.

## Author contributions

**Conceptualization:** Martin K H Ho, Tara Gomes, Muhammad Mamdani, Mina Tadrous.

**Investigation:** Martin K H Ho.

**Methodology:** Martin K H Ho, Araniy Santhireswaran.

**Project administration:** Martin K H Ho.

**Supervision:** Mina Tadrous.

**Validation:** Martin K H Ho, Mina Tadrous.

**Visualization:** Martin K H Ho, Mina Tadrous.

**Writing – original draft:** Martin K H Ho.

**Writing – review & editing:** Martin K H Ho, Araniy Santhireswaran, Tara Gomes, Muhammad Mamdani, Mina Tadrous.

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
