## [Editor Report · Decision Letter 0]

PONE-D-25-25687Comparative effectiveness and safety of insulin reference biologics versus biosimilars for types 1 and 2 diabetes mellitus: protocol for a systematic review of real-world studiesPLOS ONE

Dear Dr. Ho,

Thank you for submitting your manuscript to PLOS ONE. After careful consideration, we feel that it has merit but does not fully meet PLOS ONE’s publication criteria as it currently stands. Therefore, we invite you to submit a revised version of the manuscript that addresses the points raised during the review process.

Please Address the following:

1. Clarify the Scope of “Real-World Studies”: Provide explicit thresholds or examples based on the PRECIS-2 domains to support consistent judgment

2. Update and Expand Literature Coverage: While you mentioned that past systematic reviews are outdated (last up to 2019), it may be helpful to quantify the preliminary yield of the updated literature search and mention any especially recent high-impact real-world studies, if already identified. Consider including a table comparing key attributes of past reviews versus this planned review.

3. Risk of Bias Assessment and Heterogeneity: Given the inclusion of both observational and pragmatic RCTs, the heterogeneity could be substantial. Even if metal-analysis is not planned, outline how heterogeneity will be qualitatively assessed (e.g., using a matrix of study designs/outcomes).

4. Language: A few minor typographical and formatting issues could be improved. 

We look forward to receiving your revised manuscript.

Kind regards,

Sreedhar Dharmagadda, Ph. D.

Academic Editor

PLOS ONE

---

## [Author Response · Author response to Decision Letter 1]

29 May 2025

We thank Dr. Dharmagadda for taking the time to review our systematic review protocol. We genuinely appreciate Dr. Dharmagadda's thoughtful feedback and have made the suggested changes. We encourage the editor to review our detailed point-by-point response in the attached Response to Editor letter dated May 29, 2025 (note: not the cover letter). Below, we briefly highlight the changes:

(1) Clarifying the scope of real-world studies: The PRECIS-2 tool was originally designed to help researchers prospectively design their randomized controlled trials (RCTs). However, the PRECIS-2 authors have recognized that it can also be used to retrospectively assess pragmatism in RCTs for literature reviews. There is no simple dichotomous threshold for distinguishing pragmatic versus explanatory RCTs. It is not recommended to sum up the domain scores, rank-order the domains by importance, or set a minimum number of domains to determine pragmatism. Consistent with these recommendations, we will not apply an explicit threshold. The decision on whether an RCT is pragmatic will be guided by PRECIS-2 scores but ultimately up to the reviewers. Nevertheless, we agree that we can be more explicit to guide reviewers on how to use PRECIS-2 to inform their judgement and discussion. Furthermore, to promote transparency and reproducibility, we will publish our PRECIS-2 scores for candidate RCTs.

(2) Update and expand literature coverage: Thank you for this feedback. We agree that these changes can help highlight the novelty. We now cite several relevant studies that we came across in our feasibility search in the Introduction. However, we are cautious with the wording, as our feasibility search was only a scan of published literature and may underrepresent the growing body of evidence since 2019. Furthermore, we have added a new table to the appendix (S2 Appendix) that compares the past reviews with this planned review.

(3) Heterogeneity: Thank you for this suggestion. To qualitatively assess heterogeneity, we will present our findings in a summary table (headers: author (year), insulin product, funding source, diabetes mellitus type, number of patients, follow-up duration, relevant outcomes, authors' conclusions), sorted by study design. This summary table can illustrate whether different study designs consistently report different patient characteristics and outcomes. We have added this shell table to the appendix (S7 Appendix).

(4) Typographical and formatting issues: We identified some formatting issues and made the edits listed in the attached letter. Of note, “aspart” (drug name), “OAIster” (catalog name), “citationchaser” (R package), and “textword” (field-specific terminology) have been marked by Microsoft Word as spelling mistakes. We confirm that these are the correct spellings.

We would like to remind the editor that the above are only excerpts from our attached Response to Editor letter dated May 29, 2025. Please view our letter for the detailed changes.

---

## [Editor Report · Decision Letter 1]

Comparative effectiveness and safety of insulin reference biologics versus biosimilars for types 1 and 2 diabetes mellitus: protocol for a systematic review of real-world studies

PONE-D-25-25687R1

Dear Dr. Ho,

We’re pleased to inform you that your manuscript has been judged scientifically suitable for publication and will be formally accepted for publication once it meets all outstanding technical requirements.

Kind regards,

Sreedhar Dharmagadda, Ph. D.

Academic Editor

PLOS ONE
---

## [Editor Report · Acceptance letter]

PONE-D-25-25687R1

PLOS ONE

Dear Dr. Ho,

I'm pleased to inform you that your manuscript has been deemed suitable for publication in PLOS ONE. Congratulations! Your manuscript is now being handed over to our production team.

Kind regards,

on behalf of

Dr. Sreedhar Dharmagadda

Academic Editor

PLOS ONE